# Buccal Mucosa Cells as a Potential Diagnostic Tool to Study Onset and Progression of Arrhythmogenic Cardiomyopathy

**DOI:** 10.3390/ijms23010057

**Published:** 2021-12-21

**Authors:** Helen E. Driessen, Stephanie M. van der Voorn, Mimount Bourfiss, Freyja H. M. van Lint, Ferogh Mirzad, Laila El Onsri, Marc A. Vos, Toon A. B. van Veen

**Affiliations:** 1Department of Medical Physiology, Division Heart & Lungs, University Medical Center Utrecht, 3584 CM Utrecht, The Netherlands; h.e.driessen-4@umcutrecht.nl (H.E.D.); m.m.vandervoorn-4@umcutrecht.nl (S.M.v.d.V.); fmirzad1@hotmail.com (F.M.); 2544437@student.vu.nl (L.E.O.); m.a.vos@umcutrecht.nl (M.A.V.); 2Department of Cardiology, Division Heart & Lungs, University Medical Center Utrecht, 3508 GA Utrecht, The Netherlands; m.bourfiss-2@umcutrecht.nl; 3Department of Genetics, Division Heart & Lungs, University Medical Center Utrecht, 3584 CX Utrecht, The Netherlands; f.vanlint@amsterdamumc.nl

**Keywords:** arrhythmogenic cardiomyopathy, buccal mucosa, phospholamban, diagnosis, plakoglobin

## Abstract

In arrhythmogenic cardiomyopathy (ACM) pathogenic variants are found in genes encoding desmosomal proteins and in non-desmosomal genes, such as *phospholamban* (*PLN*, p.Arg14del variant). Previous research showed that plakoglobin protein levels and localization in the cardiac tissue of ACM patients, and *PLN* p.Arg14del patients diagnosed with an ACM phenotype, are disturbed. Moreover, the effects of pathogenic variants in desmosomal genes are reflected in non-cardiac tissues like buccal mucosa cells (BMC) which could serve as a promising new and non-invasive tool to support diagnosis. We collected the BMC of 33 ACM patients, 17 *PLN* p.Arg14del patients and 34 controls, labelled the BMC with anti-plakoglobin antibodies at different concentrations, and scored their membrane labelling. We found that plakoglobin protein levels were significantly reduced in BMC obtained from diagnosed ACM patients and preclinical variant carriers when compared to controls. This effect was independent from age and sex. Moderate to strong correlations were found with the revised 2010 Task Force Criteria score which is commonly used for ACM diagnosis (r_s_ = −0.67, *n* = 64, *p* < 0.0001 and r_s_ = −0.71, *n* = 64, *p* < 0.0001). In contrast, plakoglobin scores in *PLN* p.Arg14del patients were comparable to controls (*p* > 0.209), which suggests differences in underlying etiology. However, for the individual diagnosis of the ‘classical’ ACM patient, this method might not be discriminative enough to distinguish true patients from variant carriers and controls, because of the high interindividual variability.

## 1. Introduction

Arrhythmogenic cardiomyopathy (ACM) is a progressive genetic cardiomyopathy associated with heart failure and ventricular arrhythmias and can even present with sudden cardiac death, especially during the early (concealed) phase of the disease [1]. Inheritance is usually autosomal dominant, with an incomplete and age-dependent penetrance. Most of the pathogenic variants are found in genes encoding desmosomal proteins, most often in *plakophilin-2* (*PKP2*) [2]. These patients are commonly considered to present the ‘classical’ form of ACM [3]. However, pathogenic variants in non-desmosomal genes have also been reported. One of such a non-desmosomal variant is in the *phospholamban* gene (*PLN*, p.Arg14del), which can cause next to ACM, also a dilated cardiomyopathy (DCM) phenotype in patients [4].

Currently, diagnosis of ACM is based on the 2010 revised Task Force Criteria (TFC) [5]. The TFC include family history (genetic aberrancies), contractile dysfunction, structural alterations, cardiac tissue characteristics, and electrophysiological abnormalities [5]. Tissue characteristics include myocardial atrophy and the subsequent fibrous fatty replacement of the right ventricular (RV) free wall, for which tissue can be obtained using invasive endomyocardial biopsies. However, the specificity and sensitivity of these biopsies are highly variable and depend on the sample site, because the pathogenic remodelling of cardiac tissue during ACM has a non-uniform character [6]. Biopsies are usually taken from the septal endomyocardium which is often spared from fibrofatty replacement, resulting in a high chance of a non-informative analysis. Moreover, due to the invasive nature and risk of mortality associated with the procedure, it is not desirable or ethically acceptable, to collect endocardial biopsies from family members [7]. Therefore, other characteristics that can function as a biomarker for ACM are interesting alternatives, such as a reduction of the myocardial plakoglobin protein level, commonly observed in the ventricular tissue of both ACM and *PLN* p.Arg14del patients diagnosed with a ACM phenotype [4,8,9,10,11]. On the contrary, only one out of nine *PLN* patients diagnosed with a DCM phenotype showed this plakoglobin reduction in cardiac tissue [4]. Alterations in plakoglobin protein abundance could be a valuable additive tool to discriminate between the affected and unaffected family members of ACM patients, to discriminate between the ACM/DCM phenotype in *PLN* p.Arg14del patients, and perhaps to follow the onset and progression of the disease. Therefore, there is a demand for surrogate tissue that can function as a mirror of the heart, that can be obtained in a less invasive way, and from which we can obtain information about the plakoglobin status of subjects and predict which subjects are at risk of developing ACM, with minimal associated risks. 

Recently, buccal mucosa cells (BMC) were suggested as a promising surrogate for endomyocardial biopsies [12]. The rationale for investigating the potential of BMC originates from the existence of cardiocutaneous syndromes, like Naxos disease and Carvajal syndrome. Naxos disease is caused by a homozygous pathogenic variant in the *plakoglobin* gene (*JUP*), leading to ACM (100% penetrance), woolly hair, and palmoplantar keratoderma, [13] whereas in Carvajal syndrome, a recessive homozygous pathogenic variant in the *desmoplakin* gene (*DSP*) gives rise to similar symptoms [14,15]. These diseases show that the effects of pathogenic variants in desmosomal genes are also presented in tissues other than the heart, like BMC which do functionally express those mutated genes. Although there are currently no extracardial complications described in classical ACM patients directly related to the underlying genetic cause, this does not exclude the possibility that there are asymptomatic subtle changes present in desmosomes in non-cardiac tissues that can be unveiled by the microscopic evaluation of, as we hypothesize, BMC. 

Here we investigated the clinical usability of BMC as a diagnostic tool to classify patients at risk of developing ACM. We analyzed the plasma membrane localization of plakoglobin in human healthy control BMC and BMC obtained from ACM patients (symptomatic and preclinical carriers) and *PLN* p.Arg14del carriers (diagnosed with ACM, DCM or being preclinical).

## 2. Results

### 2.1. Patient Characteristics

From the total of 84 included subjects, Table 1 shows that our control group consisted of 34 individuals of which 14 (41%) were male and 20 (59%) were female, with an average age of 38.76 ± 2.65 years. With regard to the aspect of sex, our control group did not differ from the ACM or *PLN* p.Arg14del classified cohorts. However, there was a significant difference in average age between controls and ACM cohort (*p* = 0.004), and between controls and *PLN* cohort (*p* = 0.001). Of the 33 patients with or at risk of ACM, 19 (58%) were male and 14 (42%) were female. The average age was 49.73 ± 2.48 years. Almost all included patients displayed genetic variants in the desmosomal genes (*PKP2 n* = 28, *DSP n* = 2, and *JUP n* = 1), and two ACM patients were gene elusive. A total of 19 patients (58%) had a definite diagnosis of ACM with a TFC of ≥4, and 14 (42%) were classified as preclinical variant carriers. The average TFC score in this cohort was 4.70 ± 0.43. The cohort of 17 *PLN* p.Arg14del patients consisted of 6 (35%) males and 11 (65%) females. The average TFC score of the *PLN* p.Arg14del cohort was 1.75 ± 0.60 but only 3 patients (18%) were diagnosed with ACM (TFC of ≥4), 7 (41%) with DCM, and 7 (41%) were preclinical *PLN* p.Arg14del variant carriers. 

### 2.2. Interpretation of Buccal Mucosa Smears

In a previous study we have reported that immunohistochemistry with labelling against plakoglobin on human cardiac specimens is highly sensitive to the dilution of the antibody used [8]. Since we are unaware of the potential differences in plakoglobin protein levels in buccal mucosa and heart, we decided to apply dilution experiments in control, ACM and *PLN* p.Arg14del BMC smears. The membrane labelling of plakoglobin was scored in cell clusters consisting of at least three cells on a scale of 0 to 4, where 4 is the most abundant and clear membrane labelling (as is the example in Figure 1). At all dilutions and in all swabs, the vast majority of cells (single and clustered) had no membrane labelling. Given the continuous turnover of BMC, we hypothesized that these cells were already in the process of being shed from the buccal mucosa and replaced by younger BMC. Therefore, they could be in a phase where membrane proteins are downregulated, including the desmosomal proteins. Only samples with enough cell clusters were included to prevent the incorrect scoring of membrane labelling due to the presence of only a limited number of cells. 

### 2.3. Plakoglobin Membrane Labelling in Healthy Controls Is Not Affected by Age or Sex

Because of the difference in age between the controls and both the patients with, or at risk of, ACM as well as the *PLN* p.Arg14del patients, we investigated the potential confounding effects of age, and also sex, on plakoglobin membrane labelling in healthy controls. In all four applied dilutions, scores of plakoglobin labelling did not differ between males and females, as is depicted in Figure 2A for the ACM cohort. There was no correlation between age and plakoglobin labelling in all applied dilutions (r_s_ = 0.24, *n* = 31, *p* = 0.098 for 1:5000; r_s_ = 0.24, *n* = 32, *p* = 0.096 for 1:10,000; r_s_ = 0.29, *n* = 32, *p* = 0.053 for 1:20,000; r_s_ = 0.15, *n* = 33, *p* = 0.199 for 1:40,000), see Figure 2B.

### 2.4. Plakoglobin Membrane Labelling Is Compromised in ACM Patients

In healthy controls, plakoglobin labelling only diminished slightly in the higher dilutions of the antibody, as is illustrated in Figure 3 and the heat map in Figure 4. Some heterogeneity/variability is seen in plakoglobin labelling among the healthy controls: some individuals showed strong plakoglobin labelling even at a dilution of 1:40,000, while others showed relatively poor labelling already at a dilution of 1:10,000 (e.g., C3, C23 and C32, Figure 4). In general, in ACM patients we observed less abundant plakoglobin labelling compared to control subjects, which is visualized as the predominant red and yellow colors in the heat map of Figure 4. Next, the ACM cohort was split into preclinical variant carriers and patients with an ACM diagnosis. Both preclinical variant carriers and symptomatic ACM patients showed significantly (*p* = 0.002) decreased plakoglobin labelling compared to controls as depicted in Figure 5A. Similar to the controls, a high variability of plakoglobin scores was noticed among the ACM cohort. For example, subjects A6, A10 and A18 still showed moderate membrane labelling at a dilution of 1:40,000, whereas e.g., subjects A1, A8 and A19 showed no membrane labelling at a dilution of 1:5000. 

### 2.5. Plakoglobin Staining Moderately Correlates with 2010 TFC Score in ACM Patients

When correlating plakoglobin scores with the diagnostic 2010 TFC scores of all ACM patients, including both preclinical variant carriers and symptomatic patients, we found a moderate negative correlation with a 1:5000 dilution as is shown in Figure 5B’s upper left panel (r_s_ = −0.67, *n* = 64, *p* < 0.0001). Plakoglobin score after a dilution of 1:10,000 showed a strong correlation with 2010 TFC scores (r_s_ = −0.71, *n* = 64, *p* < 0.0001 Figure 5B upper right panel). Dilutions of 1:20,000 and 1:40,000 resulted in weaker but still significant correlations with 2010 TFC scores, as is depicted in Figure 5B’s lower left and right panels (r_s_ = −0.60, *n* = 65, *p* < 0.0001 and r_s_ = −0.55, *n* = 66, *p* < 0.0001). To further elaborate on the relation between the status of plakoglobin levels in BMC and clinical status of the ACM patients, we related the plakoglobin scores to electrophysiological data obtained from 24 h Holter recordings. Despite the fact that these clinical data were available in only 11 of our included patients, we found a strong negative correlation of plakoglobin score (1:5000 dilution) with the amount of premature ventricular contractions that occurred in 24 h; see Figure 5C (r_s_ = −0.71, *n* = 11, *p* = 0.02). 

### 2.6. Plakoglobin Labelling Is Not Suitable as Classification Tool

To test whether plakoglobin labelling is applicable in the clinic as a differentiation tool to classify patients at risk of developing ACM, we calculated the sensitivity and specificity using a plakoglobin score of 2 as cut off value. This means that a score ≥2 would classify the subject as healthy (our healthy control group) and a score of <2 would classify the subject at risk (our ACM cohort). A dilution of 1:5000 provided a low sensitivity of 0.55 and a high specificity of 0.97. For a dilution of 1:10,000 sensitivity was 0.66 and specificity 0.91, for 1:20,000 sensitivity was 0.88 and specificity 0.69 and for 1:40,000 sensitivity was 0.94 and specificity 0.58, as shown in Figure 5D. 

### 2.7. Plakoglobin Labelling in PLN p.Arg14del Patients Is Comparable to Healthy Controls

In patients with a *PLN* p.Arg14del variant, plakoglobin labelling upon all applied antibody dilutions appeared indistinctive compared to labelling in healthy controls (Figure 6 and Figure 7A). Because the *PLN* p.Arg14del cohort consisted of preclinical p.Arg14del variant carriers (*n* = 7), patients with an ACM (*n* = 3) or DCM (*n* = 7) diagnosis, we analyzed the possible differences among subgroups. Notably, these subgroups consisted of a very small number of subjects; therefore, only trends could be observed. Overall labelling in preclinical carriers and DCM patients was similar to controls, which was, however, less pronounced at a dilution 1:10,000 with respect to DCM patients (Figure 7B). Interestingly, the (only) three *PLN* p.Arg14del patients diagnosed with ACM showed a lower (median 1.50 (1–1.75) compared to 3 (2–4) controls) plakoglobin score at all dilutions (Figure 7B).

## 3. Discussion

In this study, we aimed to test the hypothesis that extracardiac tissue expressing desmosomal proteins, like BMC, could provide a non-invasive predictive diagnostic tool for ACM onset and the progression of the disease. Secondly, whether our approach would be discriminative in two different cohorts of patients: one composed of patients with ‘classical’ pathogenic variants in genes encoding desmosomal proteins and one with patients carrying a *PLN* p.Arg14del pathogenic variant. Our hypothesis is based on the fact that in two other diseases that are closely related to ACM, Carvajal syndrome and Naxos disease, pathogenic variants in genes encoding desmosomal proteins (*JUP* and *DSP*, respectively) present with pathological manifestations in the heart but also in other tissues (skin and hair) [13,14,15]. In our study, we focussed on alterations in the membrane labelling of plakoglobin in BMC. Plakoglobin is robustly expressed in BMC, and it is thought that this desmosomal protein is generally affected in the cardiac specimens of nearly all ACM patients regardless of their underlying pathogenic variant, including those with *PLN* p.Arg14del [8,9,11,16]. Moreover, we previously showed that, when aberrant plakoglobin labelling in the cardiac tissue of ACM patients was identified, this was always accompanied by the diminished presence of Nav1.5 (the cardiac sodium channel) and/or Cx43 (the predominant gap junction protein in the ventricles) [10,17]. This fact relates to disturbances in plakoglobin and to potential alterations in excitability and impulse propagation, thereby adhering to the arrhythmogenic character of the disease [17,18,19].

Moreover, in a study of Asimaki et al. [12], plakoglobin showed the most consistent changes in the buccal mucosa smears of ACM patients. In our current study, we performed dilution experiments in control subjects and patients to investigate the diagnostic potential of plakoglobin labelling in the BMC of ACM patients. In line with the data presented in the study of Asimaki [12], we confirm that, in the buccal mucosa swabs of ACM patients, significantly less plakoglobin is present compared to healthy controls and that there is a moderate though significant correlation with disease severity as correlated with the TFC scores and the amount of PVCs as derived from 24 h Holter recordings. These results were independent of sex and age. However, it should be noted that there was a substantial variation in sensitivity and specificity and a high inter-individual variability in plakoglobin labelling in controls and ACM patients. This compromises the use of this labelling approach to correctly discriminate between affected and unaffected subjects or to use it as a prospective measure to follow disease onset and progression. Indeed, when we subcategorized diagnosed ACM patients and preclinical variant carriers, both groups showed a significant reduction in plakoglobin protein scores when compared to controls, without being different between those two subgroups. Moreover, the vast number of collected BMC without membrane labelling in the samples, probably representing cells that are at the end of the shedding process and the most likely ones to be released easily during swapping, is a significant limitation with regard to clinical implementation.

Whether the diminished signals in BMC align with pro-arrhythmic remodelling is difficult to predict in ACM patients. BMC do express Cx43 but in very limited amounts, and we have not been able to make any reliable observation of potential changes in the membrane presence of Cx43. Nav1.5, the other factor that previously appeared to be co-regulated with changes in plakoglobin signal, is not expressed in BMC. Interestingly, our data revealed relevant and significant correlations between plakoglobin protein reduction in BMC and 2010 TFC scores; the most severely affected ACM patients (highest TFC scores) showed the largest reduction in plakoglobin. Moreover, poor plakoglobin scores correlated significantly with the number of PVCs. This suggests that plakoglobin scores may have prognostic value regarding arrhythmogenic risk prediction. Obviously, a direct comparison between BMC and cardiac specimen obtained from a patient at the same time is not possible given the ethical considerations and clinical risk of taking a biopsy for research purposes. Only one additional study directly compared the immune labelling of BMC smears with cardiac tissue. This study of Begay et al. [20] investigated two patients with a filament C truncation that caused an arrhythmogenic DCM phenotype. The immuno-labelling of BMC smears and heart biopsy showed similar aberrancies, as both tissues showed a reduction in the levels of desmoplakin and SAP97 [20]. To be able to study the direct comparison of BMC and cardiac tissue further, we might have to take advantage of currently emerging mouse models genetically engineered to express patient-specific mutated proteins causative for the development of ACM.

In contradiction to the ACM cohort, within the *PLN* p.Arg14del cohort, there was no difference in the BMC levels of plakoglobin when compared to controls. Although observed in a very tiny subgroup of only three patients, our results suggest that p.Arg14del patients diagnosed with an ACM phenotype could more resemble the ‘classical’ ACM patients with a desmosomal pathogenic variant with regard to BMC plakoglobin levels. This could indicate that a p.Arg14del pathogenic variant can cause a cardiomyopathy via at least two different mechanisms, both with calcium as a key player, leading to either global structural changes causing DCM or via decreased desmosomal protein expression causing an ACM phenotype.

## 4. Materials and Methods

### 4.1. Study Population and Clinical Data Assessment

Subjects were invited to participate in this study via a letter after application for a patient day for ACM patients or carriers of a *PLN* p.Arg14del pathogenic variant. Controls were selected from unaffected family members or healthy colleagues from our medical center. In total, 84 subjects were enrolled in this study and divided into three groups: 34 healthy controls (14 males), 33 ACM patients (preclinical carriers of desmosomal gene variants and ACM diagnosed, 19 males) and 17 p.Arg14del patients (diagnosed with ACM, DCM or being preclinical carriers, 6 males). ACM patients and p.Arg14del patients were defined as preclinical variant carriers (TFC < 4), or as being diagnosed with ACM (TFC ≥ 4) with or without a known pathogenic variant. In the *PLN* cohort, patients were categorized as DCM when adhering to the Henry criteria [21]. Controls were defined as individuals without any history of cardiac disease. All subjects provided written informed consent. Sample collection protocols and study design were approved by the local Biobank committee (protocol number TCBio 14-513), an ethical review board of the University Medical Center Utrecht. After inclusion of the subjects, clinical data were assessed from the ACM and *PLN* Registry as hosted in REDCap. These prospective registries collect, amongst others, clinical data, demographics, symptoms, medication use, family history and genetic analysis from Dutch ACM and *PLN* patients [22]. From this registry, Holter recordings were also extracted, which were performed a maximum of two years prior to, or following sample collection, to study a potential correlation between BMC scores and pro-arrhythmic remodelling.

### 4.2. Collection and Sample Preparation

A cotton-tipped swab was used to collect BMC from the inside of the cheeks of the participants. Each side was swabbed for 30 s to allow for sufficient cells to adhere to the cotton top of the swab. Subsequently, smears were made by rolling the cotton swab directly on a glass slide, and cells were immediately fixated with 70% ethanol. Samples were allowed to air dry, after which (immuno)cytochemical procedures were performed at the same day. 

### 4.3. Immunohistochemistry and H&E 

Human BMC for H&E staining were fixed with 4% paraformaldehyde (PFA), after which hematoxylin and eosin staining (Merck, Darmstadt, Germany) was performed. For immunocytochemistry experiments, BMC were fixed with 70% ethanol. After ethanol fixation, samples were washed with phosphate buffered saline (PBS) and incubated in blocking solution (3% normal goat serum, 1% bovine serum albumin, 0.15% TritonX-100 in PBS) for 45 min [12]. After washing with 1% Triton X-100 in PBS, the specimens were incubated with the primary antibody (mouse monoclonal anti-plakoglobin (Merck, Darmstadt, Germany) overnight at 4 °C. The next day, samples were allowed to reach room temperature before washing again with 1% TritonX-100 in PBS, subsequently followed by a secondary antibody incubation for two hours at room temperature. The secondary antibody used was Alexa Fluor 594 (1:250) conjugated to anti-mouse whole IgG antibodies (Jackson ImmunoResearch Europe, Newmarket, UK). Sections were analyzed with a Nikon Eclipse 80i epifluorescence microscope. Pictures were taken with a Nikon Digital sight DS-2MBWc camera and NIS Elements BR 3.0 software. Membrane labelling in human buccal mucosal swabs was assessed in cell clusters of more than three cells by two independent observers in a blinded fashion. 

### 4.4. Statistical Analysis

Data are shown as median [interquartile range]. Statistics were performed by ANOVA with a Dunnet post-hoc test, Mann-Whitney test, Pearson correlation or Spearman’s correlation. Differences or correlations were considered significant if *p* < 0.05; correlations were considered weak between 0.10 and 0.40, moderate between 0.40 and 0.70, strong between 0.70 and 0.80 and very strong between 0.80 and 1.00. All analyses were performed using GraphPad Prism 9.0 (GraphPad Software, La Jolla, CA, USA). 

## 5. Conclusions

In conclusion, the results in this study confirm that, at the population level, there is a significant difference in plakoglobin labelling in the BMC membranes of diagnosed ACM patients and preclinical variant carriers compared to control subjects, with a moderate correlation to disease severity. This effect was independent from age and sex, but importantly, moderate to strong correlations were found with the revised 2010 TFC scores. However, due to the high variability within each group and the difficulty of interpreting the individual samples in a one-on-one comparison, this method is most likely not suitable to distinguish between ACM patients or healthy controls at the individual level. Given the fact that both diagnosed ACM patients and preclinical variant carriers showed similar reductions, at this moment, this approach does also not allow clinical applicability as a tool to predict disease onset. In order to do so, a much larger amount of borderline preclinical variant carriers (TFC 2-3) should be studied in comparison to those being diagnosed with ACM having high TFC scores.

## Figures and Tables

**Figure 1 ijms-23-00057-f001:**
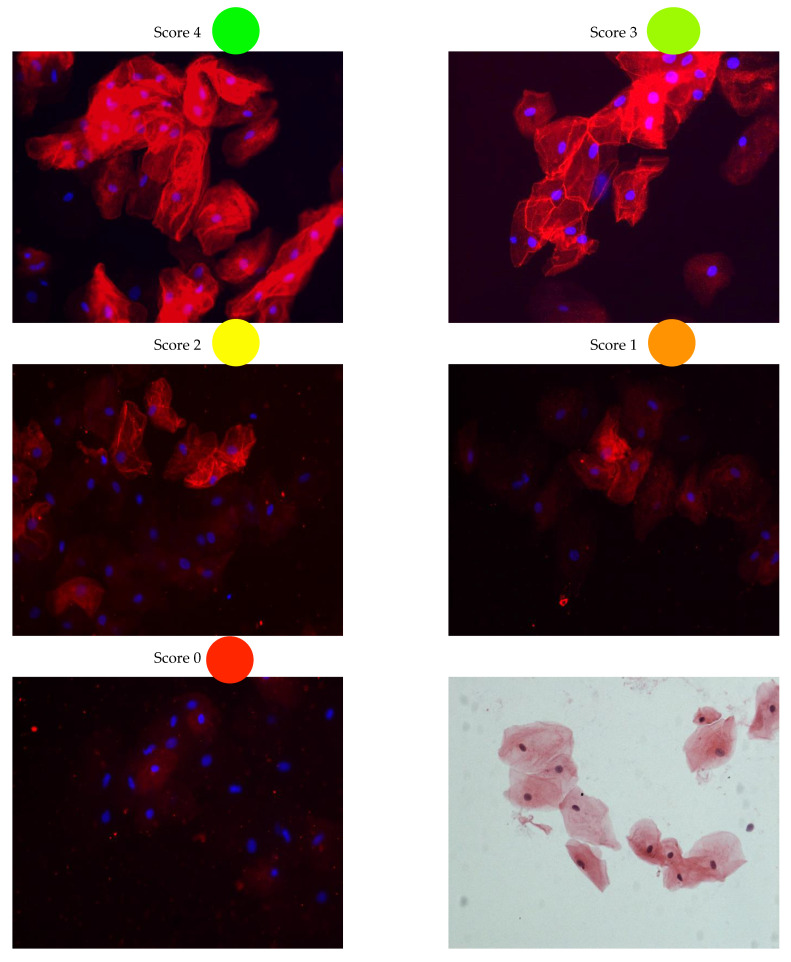
Examples of buccal mucosa scoring. A score of 4 means extreme strong membrane labelling and a score of 0 means no labelling at all. Scoring was performed on clusters consisting of more than three cells. Right bottom panel depicts a H&E staining of buccal mucosa cells. H&E; hematoxylin and eosin staining.

**Figure 2 ijms-23-00057-f002:**
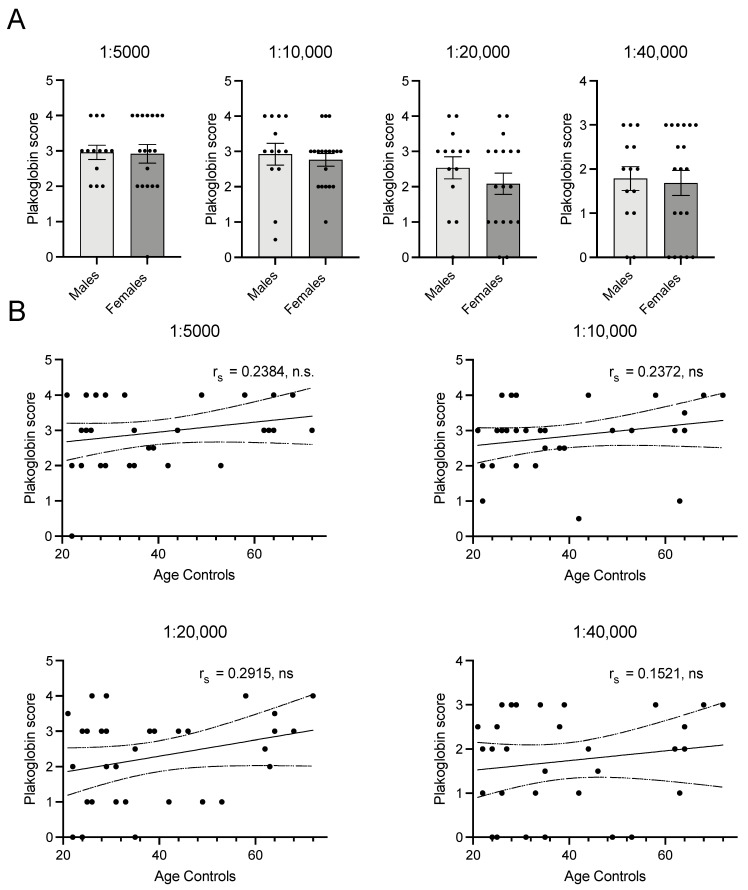
(**A**) Panel shows that there is no difference between membrane labelling score between males (*n* = 14) and females (*n* = 20) in any of the dilutions. (**B**) Depicts correlations between age and plakoglobin score. None of the dilutions show significant or biological relevant correlations with age. 1:5000 *n* = 31, 1:10,000 *n* = 32, 1:20,000 *n* = 32, 1:40,000 *n* = 33. r_s_; Spearman’s rho, n.s.: not significant. • indicate individual(s) for the correlation.

**Figure 3 ijms-23-00057-f003:**
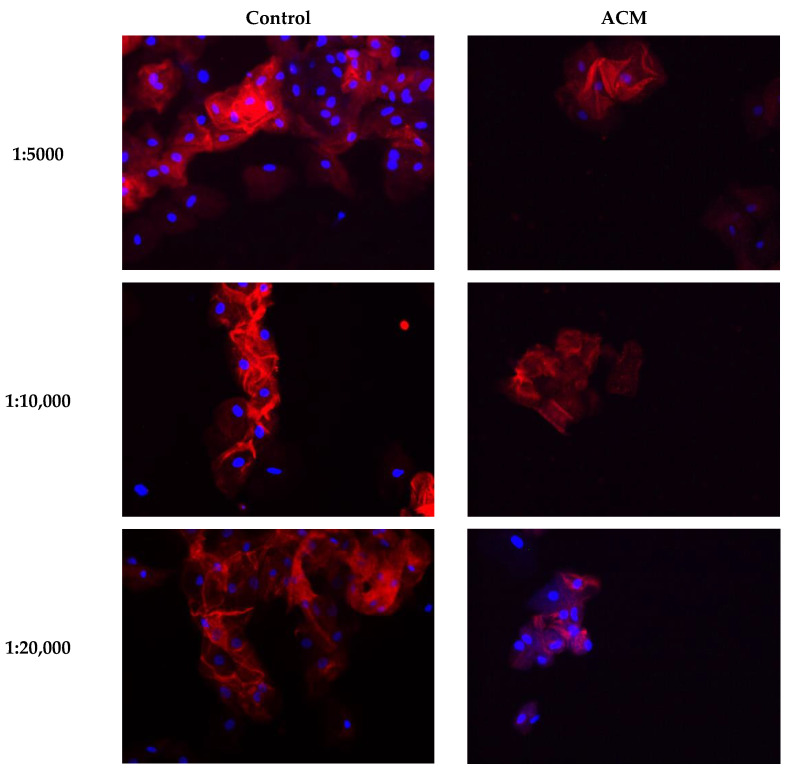
Examples of buccal mucosa smears of healthy controls and ACM patients labeled for plakoglobin at different dilutions. ACM; arrhythmogenic cardiomyopathy.

**Figure 4 ijms-23-00057-f004:**
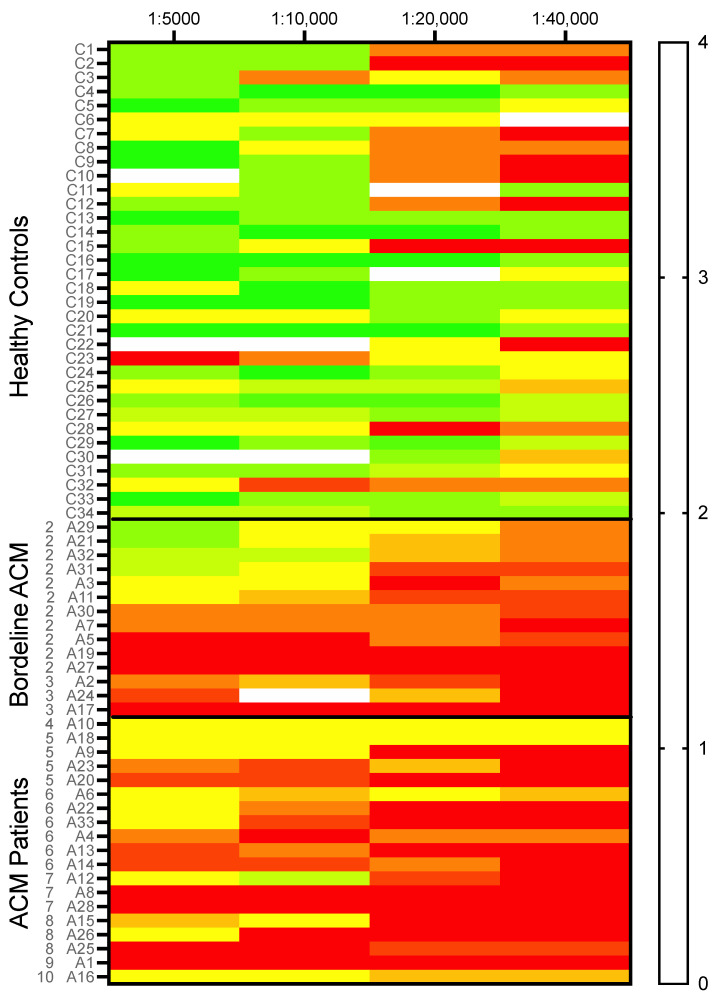
A heat map visualizing the plakoglobin score of all individual subjects (healthy controls *n* = 34, borderline ACM *n* = 14 and ACM diagnosed patients *n* = 19) for all four dilutions. A clear pattern is visible where healthy controls in general tend to show higher scores than borderline and ACM-diagnosed patients. Colors correspond to the score as shown in the right panel. At the left axis, controls (C) and ACM patient (A) numbers are indicated. Furthermore, for each ACM patient their TFC score is listed. ACM; arrhythmogenic cardiomyopathy.

**Figure 5 ijms-23-00057-f005:**
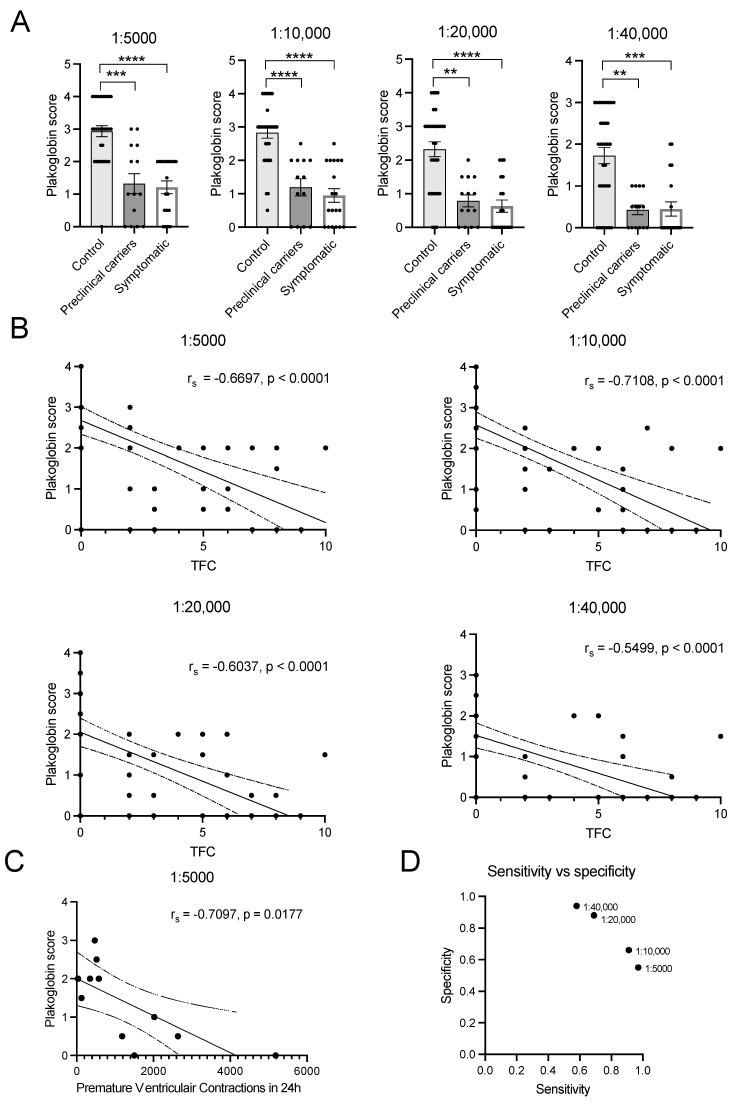
(**A**) Strong and significant differences at the population level between healthy controls (*n* = 34), preclinical carriers (*n* = 14) and ACM diagnosed patients (*n* = 19) in all four dilutions. (**B**) Spearman correlation graphs for plakoglobin score and TFC are shown. (**C**) There is a strong significant correlation with premature ventricular contractions in ACM patients (*n* = 11). (**D**) Comparison of the sensitivity and specificity score of each dilution used in this study. TFC; Task Force Criteria score 2010, r_s_; Spearman’s rho. • indicate individual(s) for the correlation or represent dilution factor in figure D. **** *p* < 0.0001, *** *p* < 0.001, ** *p* < 0.01.

**Figure 6 ijms-23-00057-f006:**
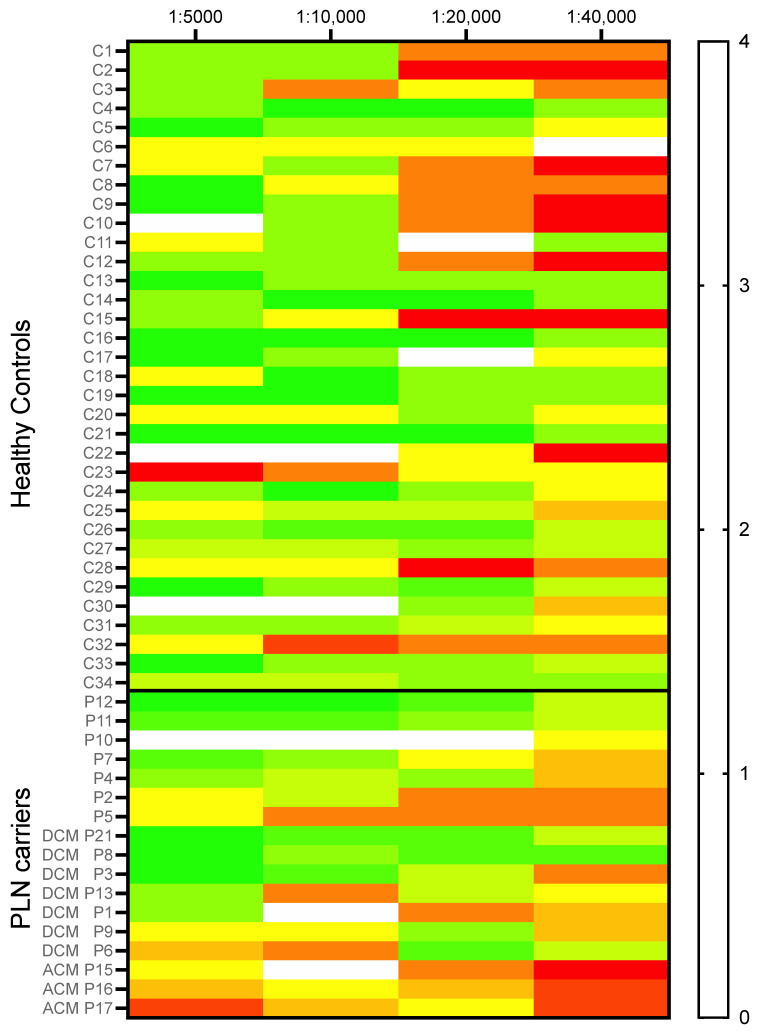
A heat map visualizing plakoglobin scores of all individual healthy controls (*n* = 34) and *PLN* carriers (*n* = 17), for all four dilutions. No clear difference was found between controls and *PLN* carriers in plakoglobin score. Colors correspond to score as shown in the right panel. At the left axis, controls (C) and *PLN* p.Arg14del patient (P) numbers are indicated. Furthermore, for *PLN* patients their clinical diagnosis is indicated when specified. PLN; phospholamban, ACM; arrhythmogenic cardiomyopathy, DCM; dilated cardiomyopathy.

**Figure 7 ijms-23-00057-f007:**
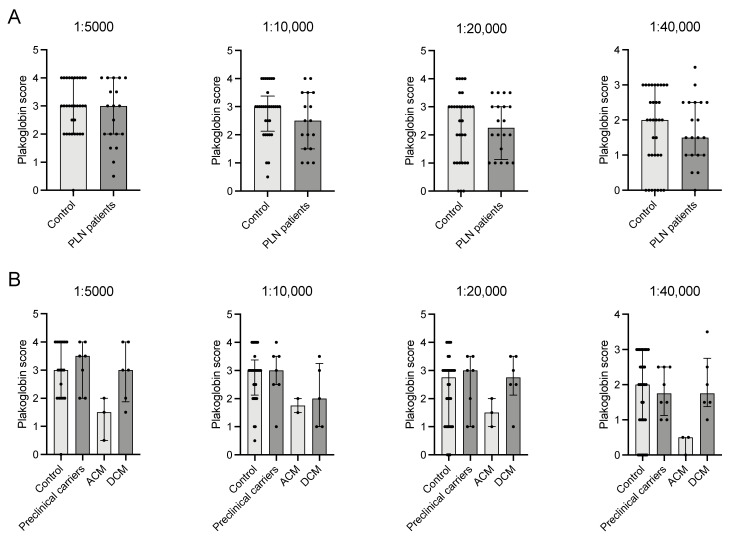
(**A**) No significant differences at the population level between healthy controls (*n* = 34) and *PLN* carriers (*n* = 17) in any of the dilutions. (**B**) Graphs depict differences in plakoglobin score between controls (*n* = 34), *PLN* variant carriers that are preclinical (*n* = 7), with an ACM (*n* = 3) or DCM (*n* = 7) phenotype. ACM *PLN* patients seem to adhere more to the ACM patients’ pattern of plakoglobin score. PLN; phospholamban, ACM; arrhythmogenic cardiomyopathy, DCM; dilated cardiomyopathy. • indicate individual(s) for the correlation.

**Table 1 ijms-23-00057-t001:** Table shows patient characteristics of the ACM and *PLN* cohort. Data depicted as average ±standard error of the mean. PKP2; plakophilin 2, DSP; desmoplakin, PLN; phospholamban, TFC Task Force Criteria 2010, ACM; arrhythmogenic cardiomyopathy, DCM; dilated cardiomyopathy. Students t-test when appropriate, *** *p* < 0.001, ** *p* < 0.01.

	Controls	ACM	*PLN*
General			
*n*	34	33	17
*A* *verage age*	38.76 (±2.65)	49.73 (±2.48) **	54.29 (±3.21) ***
*M* *ale*	14 (41)	19 (58)	6 (35)
Pathogenic variant			
*PKP2*	-	28 (85)	-
*DSP*	-	2 (6)	-
*P* *lakoglobin*	-	1 (3)	-
*PLN*	-	-	17 (100)
*N* *o known pathogenic variant*	-	2 (6)	-
TFC			
*Average*	-	4.70 (±0.43)	1.75 (±0.60)
*ACM (TFC ≥ 4)*	-	19 (58)	3 (18)
*DCM*	-	-	7 (41)
*P* *reclinical variant carrier*	-	14 (42)	7 (41)

## Data Availability

The data presented in this study are available on request from the corresponding author.

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
