# Peer review of "Buccal Mucosa Cells as a Potential Diagnostic Tool to Study Onset and Progression of Arrhythmogenic Cardiomyopathy"

_ijms, 2021, doi:10.3390/ijms23010057_

Round 1

Reviewer 1 Report

The authors presented very interesting study aimed to find simple approach for defining patients at risk of Arrhythmogenic cardiomyopathy. They used buccal mucosa cells to evaluate the level of plakoglobin in tissue samples. They found that the level of plakoglobin was lower in samples of patients with ARVC and carries of mutation in desmosomal genes compared to patients having mutation in non-desmosomal genes (phospholamban) and controls. Also, they found the  correlation between the level of plakoglobin and the severity of ARVC. The week side of the study is small number of groups. 

I have a few comments:

1) Could the authors describe the study cohort and selection of study subjects. As I understand the authors used the data from a register. There is no description of the register in the text. Also, it is not clear how the controls were selected and from where.

2) I did not find the Conclusion. It seems to me that it is worth shortly summarize the main findings.

Author Response

We would like to thank the reviewers for their efforts and constructive advice regarding our manuscript. Below, we have addressed the reviewer’s comments in a point-wise fashion and revised the manuscript accordingly (track changed). We believe by implementing the obtained suggestions the manuscript has been significantly improved.

Reviewer 1:

The authors presented very interesting study aimed to find simple approach for defining patients at risk of Arrhythmogenic cardiomyopathy. They used buccal mucosa cells to evaluate the level of plakoglobin in tissue samples. They found that the level of plakoglobin was lower in samples of patients with ARVC and carries of mutation in desmosomal genes compared to patients having mutation in non-desmosomal genes (phospholamban) and controls. Also, they found the correlation between the level of plakoglobin and the severity of ARVC. The week side of the study is small number of groups.

Point 1: Could the authors describe the study cohort and selection of study subjects. As I understand the authors used the data from a register. There is no description of the register in the text. Also, it is not clear how the controls were selected and from where.

This information was, and is, included in section 4 Materials and Methods, subheading ‘Study population and clinical data assessment’ (lines 323-344). We also included information about the register in more detail and added information how we selected our study subjects and controls. To specify, we first selected the subjects on specific patient days where ACM and PLN patients come together with clinicians and researchers that are actively involved in their disease. During those days the patients are further informed about the progression in treatment and research of their disease, and are also able to participate in research on a voluntary basis upon written informed consent. After collection of the BMC and immunolabeling, we collected the respective clinical data of the patients (in an anonymous fashion) from the Redcap database with permission of the subjects. This database includes all collected clinical data from these patients and is contiously updated when new data become available. Controls were unaffected family members that also participated during those days, and healthy colleagues from our medical center. 

Point 2: I did not find the Conclusion. It seems to me that it is worth shortly summarize the main findings.

We added a section with conclusions after the section ‘Materials and Methods’ to summarize our main findings (lines 405-415).

Reviewer 2 Report

The authors collected buccal mucosa cells (BMC) of 33 arrhythmogenic cardiomyopathy (ACM) patients, 17 PLN p.Arg14del patients and 34 controls, labelled the BMC with anti-plakoglobin antibodies at different concentrations, and scored their membrane labeling. ACM patients and p.Arg14del patients were defined as preclinical variant carriers (TFC <4), or as being diagnosed with ACM (TFC >4) with or without a known pathogenic genetic variant. In the PLN cohort, patients were categorized as DCM when adhering to the Henry criteria. They observed that plakoglobin protein levels were significantly reduced in BMC obtained from diagnosed ACM patients and preclinical variant carriers when compared to controls. Moderate to strong correlations were found with the revised 2010 Task Force Criteria Score which is commonly used for ACM diagnosis. In contrast, plakoglobin scores in PLN 27p.Arg14del patients were comparable to controls suggesting differences in underlying etiology. They concluded that for individual diagnosis of the ‘classical’ ACM patient, this method might not be discriminative enough to distinguish true patients from variant carriers and controls, because of the high interindividual variability.

Comments/queries

  1. In the era of ‘-omic’ technologies and precision medicine ACM serves as a prime example of a heterogeneous family of disorders in which there are multiple genetic and non-genetic causes. Who is the classical ACM patient?
  2. Changes in desmosomal protein expression, may be determined already during the early phase of ACM. Although cardiac biopsies may provide this information, they are not easily obtained. Hence, new biomarkers are required and a tentatively promising approach comprises the use of BMC to study ACM pathogenicity and disease progression. The idea to use such specimen is based on the fact that patients with Carvajal syndrome or Naxos disease (having mutations in DSP and JUP,respectively) present with cardiomyopathies reminiscent of ACM but also with extra-cardiac features in tissues that also express these desmosomal proteins. Following this line of thinking, it was demonstrated that expression of desmosomal proteins including plakoglobin is altered in BMC from ACM patients.
  3. A recent study employing analysis of serial buccal mucosa samples of children and adolescents from families with known ACM-causing variants reported that junctional protein re-localization does not correlate with the presence of an ACM-causing variant but instead correlates with the onset of disease. No changes were seen in buccal smears until there was clinical evidence of disease. In addition, progressive shifts in the distribution of some proteins correlated with worsening of the disease phenotype. Further, restoration of junctional signal for Cx43, the major ventricular gap junction protein, in a patient with a favourable response to anti-arrhythmic therapy was observed. Any comments?

Author Response

We would like to thank the reviewers for their efforts and constructive advice regarding our manuscript. Below, we have addressed the reviewer’s comments in a point-wise fashion and revised the manuscript accordingly (track changed). We believe by implementing the obtained suggestions the manuscript has been significantly improved.

The authors collected buccal mucosa cells (BMC) of 33 arrhythmogenic cardiomyopathy (ACM) patients, 17 PLN p.Arg14del patients and 34 controls, labelled the BMC with anti-plakoglobin antibodies at different concentrations, and scored their membrane labeling. ACM patients and p.Arg14del patients were defined as preclinical variant carriers (TFC <4), or as being diagnosed with ACM (TFC >4) with or without a known pathogenic genetic variant. In the PLN cohort, patients were categorized as DCM when adhering to the Henry criteria. They observed that plakoglobin protein levels were significantly reduced in BMC obtained from diagnosed ACM patients and preclinical variant carriers when compared to controls. Moderate to strong correlations were found with the revised 2010 Task Force Criteria Score which is commonly used for ACM diagnosis. In contrast, plakoglobin scores in PLN 27p.Arg14del patients were comparable to controls suggesting differences in underlying etiology. They concluded that for individual diagnosis of the ‘classical’ ACM patient, this method might not be discriminative enough to distinguish true patients from variant carriers and controls, because of the high interindividual variability.

Point 1: In the era of ‘-omic’ technologies and precision medicine ACM serves as a prime example of a heterogeneous family of disorders in which there are multiple genetic and non-genetic causes. Who is the classical ACM patient?

We based the term ‘classical’ ACM to what was historically considered to be causative for the ACM phenotype, namely a pathogenic variant in genes encoding desmosomal proteins (plakophilin-2, desmoplakin, plakoglobin, desmoglein-2 and desmocollin-2), see PMID 31210398. This is the group of patients that we included into the ACM cohort. More recently, it was found that also non-desmosomal pathogenic variants can lead to arrhythmogenic cardiomyopathy. Therefore, we also focused on the PLN p.Arg14del pathogenic variant, as it is a founder variant in the Netherlands. We specified the term ‘classical ACM’ in the introduction on page 1, line 42 and included an additional reference to this.

Point 2: Changes in desmosomal protein expression, may be determined already during the early phase of ACM. Although cardiac biopsies may provide this information, they are not easily obtained. Hence, new biomarkers are required and a tentatively promising approach comprises the use of BMC to study ACM pathogenicity and disease progression. The idea to use such specimen is based on the fact that patients with Carvajal syndrome or Naxos disease (having mutations in DSP and JUP,respectively) present with cardiomyopathies reminiscent of ACM but also with extra-cardiac features in tissues that also express these desmosomal proteins. Following this line of thinking, it was demonstrated that expression of desmosomal proteins including plakoglobin is altered in BMC from ACM patients.

We thank the reviewer for recognizing the potential of extracardiac BMC as study objective in ACM. Since this resume does not contain a question, we are not able to comment rather than to confirm that the above description regarding the basis of our work is correct.

Point 3: A recent study employing analysis of serial buccal mucosa samples of children and adolescents from families with known ACM-causing variants reported that junctional protein re-localization does not correlate with the presence of an ACM-causing variant but instead correlates with the onset of disease. No changes were seen in buccal smears until there was clinical evidence of disease. In addition, progressive shifts in the distribution of some proteins correlated with worsening of the disease phenotype. Further, restoration of junctional signal for Cx43, the major ventricular gap junction protein, in a patient with a favourable response to anti-arrhythmic therapy was observed. Any comments?

This interesting and very recent study of Beti et al. shows that clinical manifestations were needed to detect changes in buccal mucosa smears of children (n=12, age 3-18 years). In children who were carriers of a pathogenic variant, but did not show clinical manifestations, no changes in buccal mucosa smears were detected. Changes in buccal mucosa smears correlated to clinical onset of the disease. So, this study showed that buccal mucosa might be a useful biomarker to distinguish between juvenile carriers without clinical manifestations (n=6) and patients who show clinical manifestations (n=6), although numbers are very small and pathogenic variants were rather different.

In our study, we found that in adults already patients at risk showed significant lower plakoglobin expression. Furthermore, we calculated the sensitivity and specificity of this biomarker in our cohort. Only on population level we could find differences in plakoglobin score, but on individual level, there was quite some heterogeneity to allow a discriminative diagnosis. The children study showed minor differences in buccal mucosa pattern compared to the adult study, which was performed before (Asimaki et al, Circulation A&E, 2016), as downregulation of proteins was also found in adults without clinical manifestation. This is similar to our findings in the adult patient group, where we already found downregulation of plakoglobin expression in preclinical carriers. In addition, in this recent study of Beti et al, the authors seem to find a correlation between changes in protein localization and clinical expression of the disease, likewise we found in our study where TFC score was negatively correlated to the plakoglobin score.  

Round 2

Reviewer 2 Report

I have no further comments.